# Soluble RAGE as a Prognostic Marker of Worsening in Patients Admitted to the ICU for COVID-19 Pneumonia: A Prospective Cohort Study

**DOI:** 10.3390/jcm11154571

**Published:** 2022-08-05

**Authors:** Emmanuel Besnier, Valéry Brunel, Caroline Thill, Perrine Leprêtre, Jérémy Bellien, Zoe Demailly, Sylvanie Renet, Fabienne Tamion, Thomas Clavier

**Affiliations:** 1Department of Anesthesiology and Critical Care, Rouen University Hospital, UNIROUEN, INSERM U1096, Normandie Université, F-76000 Rouen, France; 2Rouen University Hospital, INSERM CIC-CRB 1404, F-76000 Rouen, France; 3Department of General Biochemistry, Rouen University Hospital, F-76000 Rouen, France; 4Department of Biostatistics, Rouen University Hospital, F-76000 Rouen, France; 5Department of Pharmacology, Rouen University Hospital, UNIROUEN, INSERM U1096, Normandie Université, F-76000 Rouen, France; 6Medical Intensive Care Unit, Rouen University Hospital, UNIROUEN, INSERM U1096, Normandie Université, F-76000 Rouen, France; 7UNIROUEN, INSERM U1096, Normandie Université, F-76000 Rouen, France

**Keywords:** SARS-CoV-2, COVID-19, sRAGE, GRP78, unfolded protein response, endoplasmic stress response, VEGF-A

## Abstract

Background: The different waves of SARS-CoV-2 infection have strained hospital resources and, notably, intensive care units (ICUs). Identifying patients at risk of developing a critical condition is essential to correctly refer patients to the appropriate structure and to spare limited resources. The soluble form of RAGE (sRAGE), the endoplasmic stress response and its surrogates, GRP78 and VEGF-A, may be interesting markers. Methods: This was a prospective monocenter cohort study of adult patients admitted to the ICU for severe COVID-19 pneumonia. The plasma levels of sRAGE, GRP78 and VEGF-A were measured within the first 24 h. Patients were classified as critical if they further needed vasopressor therapy, renal replacement therapy, or invasive mechanical ventilation, or died during their ICU stay, and were otherwise classified as not critical. Results: A total of 98 patients were included and 39 developed a critical condition. Critical patients presented higher sRAGE (626 [450–1043] vs. 227 [137–404] pg/mL, *p* < 0.0001), interleukin-6 (43 [15–112] vs. 11 [5–20] pg/mL, *p* < 0.0001), troponin T (17 [9–39] vs. 10 [6–18] pg/mL, *p* = 0.003) and NT-pro-BNP (321 [118–446] vs. 169 [63–366] pg/mL, *p* = 0.009) plasma levels. No difference was observed for VEGF-A and GRP78. The variables independently associated with worsening in the ICU were sRAGE (1.03 [1.01–1.05] per 10 pg/mL) and age (1.7 [1.2–2.4] per 5 years). An sRAGE value of 449.5 pg/mL predicted worsening with a sensitivity of 77% and a specificity of 80%. Conclusion: sRAGE may allow the identification of patients at risk of developing a critical form of COVID-19 pneumonia, and thus may be useful to correctly refer patients to the appropriate structure of care.

## 1. Introduction

Since the first identification of SARS-CoV-2 infection in humans, the subsequent COVID-19 pandemic has been responsible for at least half a billion cases worldwide, with an estimated mortality of at least 6 million people (data from the 3 July 2022).

Many efforts have been focused on understanding the mechanisms underlying virus entry into the host system and its pathogenesis, but also to understand the pathophysiological changes. Dysregulated inflammation may participate in the pathological systemic changes observed during COVID-19, triggering endothelium dysfunction, coagulation disorders, and eventually leading to shock [1,2,3]. High levels of cytokines, such as interleukin-6 (IL-6), are associated with the severity of the pathology and poor prognosis, and likely explain the beneficial effects of dexamethasone in the most seriously ill patients [1,4]. Nevertheless, the use of IL-6 or CRP as prognostic markers seems inconsistent during COVID-19. In a recent study, Picod et al. demonstrated an association between IL-6 and mortality or organ support, but the marker’s performance in predicting mortality appeared limited, questioning its usefulness in selecting the most at-risk patients [5]. Several hypotheses may explain this lack of performance, including the pharmacokinetics of this interleukin, with an early peak of synthesis, whereas COVID-19 presents a rather slow evolution, with several days between the first symptoms and the admission to the ICU. Another explanation may be the possible phenotypic variations between individuals concerning the production of the natural antagonist, the soluble receptor of IL-6 [6]. A strategy to overcome the limitations of the classical inflammatory biomarkers may be to focus on indirect biomarkers of inflammation, i.e., pathways activated by primary inflammation and thus more likely to last over time. A better understanding of the relationship between such biomarkers and clinical outcomes might help identify patients at the highest risk of clinical deterioration.

The receptor for advanced glycation-end products (RAGE) is implicated in several pathways of inflammatory processes, notably because of its ability to directly or indirectly interact with different types of ligands [7]. Thus, RAGE acts as a major component of acute inflammation, mediated by both pathogen-associated molecular patterns (PAMPs), such as lipopolysaccharides, and damage-associated molecular patterns (DAMPs), such as the high-mobility group box 1 protein (HMGB1) [8,9]. RAGE is preferentially expressed in the type 1 alveolar epithelial cells of the lungs, and at lower levels in the skin. It may also be expressed in various types of tissue during aging, and in cases of chronic disease, such as diabetes mellitus, obesity, atherosclerosis, and some nephropathies [7,10], some of which are known to be associated with worse outcomes during SARS-CoV-2 infection.

The endoplasmic reticulum is responsible for protein regulation, owing to specific chaperones such as the 78 kDa glucose-regulated protein (GRP78), which participates in the correct folding of proteins [11]. Inflammatory settings can alter the correct endoplasmic folding, a situation known as endoplasmic reticulum stress (ERS). ERS has been described to be associated with organ failure in patients suffering from a systemic inflammation response during septic shock or cardiac surgery with cardiopulmonary bypass [12,13]. SARS-CoV-2 infection and replication within the host cell may compete with the normal folding process, and may be responsible for both GRP78 overexpression and an increased unfolded protein response [14,15,16,17]. However, there are currently no data describing the GRP78 level in COVID-19 ICU patients.

The vascular endothelial growth factor type A is a key actor implicated in the homeostasis of the microcirculation, by regulating capillary permeability, vascular tone and cellular interactions [18,19]. Pro-inflammatory cytokines induced a rise in VEGF production, notably through the indirect effect of JAK/STAT3 and MAPK pathways, with elevated plasma levels observed in septic patients [20,21].

Because inflammation is associated with higher mortality in COVID-19 disease, the identification of surrogate biomarkers may be of interest to identify patients at risk of worsening, notably during acute epidemic waves when critical care resources may be strained by a massive inflow of critically ill patients. In a large cohort, we hypothesize that plasma levels of soluble RAGE (sRAGE), GRP78, and VEGF-A, three markers of pathways activated by primary inflammation, may be associated with worse outcomes for COVID-19 ICU patients, and thus may be used as reliable prognostic biomarkers.

## 2. Materials and Methods

### 2.1. Study Design

We conducted a prospective monocenter cohort study in three intensive care units of a university hospital. This cohort was approved by an ethics committee on the 7 April 2020 (Comité de Protection des Personnes Est-III, approval number 2020-A00885-34). According to the French law, verbal approval was required from the patient or their relatives. The study was conducted in accordance with the principles of the Declaration of Helsinki related to human research. The elaboration of the manuscript was in accordance with the STROBE statement.

This cohort was described in a previous publication [3]. Briefly, patients were eligible for inclusion if they presented a “severe” condition related to a documented infection of SARS-CoV-2 (determined by polymerase chain reaction assay), defined as the need for ICU admission because of acute respiratory failure. Non-inclusion criteria were pregnancy, a documented bacterial co-infection, known limitations in life support because of patient choice or important comorbidities, and an expected death within 24 h.

Clinical data were collected at admission and during the ICU stay, including conventional characteristics, specific treatments for COVID-19 disease (anticoagulation, corticosteroids, and others), Simplified Acute Physiology Score II (SAPS-II) and Sequential Organ Failure Assessment (SOFA) scores, use of non-invasive ventilation (NIV) or high-flow nasal cannula (HFNC) therapies and the subsequent ROX index (defined as $\mathrm{ROX}=\frac{{\mathrm{SpO}}_{2}/{\mathrm{FiO}}_{2}}{Respiratory\;Rate}$) and PaO_2_/FiO_2_ ratio. Conventional biological characteristics included blood count, urea, creatinine, C-reactive protein (CRP), lactatemia, high-sensitivity troponin T, and N-terminal pro-brain natriuretic peptide (NT-pro-BNP). Patients were followed for up to 28 days or until death.

### 2.2. Endpoints

The primary endpoint was the clinical worsening from a “severe” towards a “critical” condition within 28 days after ICU admission, according to NMA-COVID Initiative Network definitions [22]. According to the NIH COVID-19 treatment guidelines [23], a patient was defined as “critical” if they presented at least one of the following outcomes: (1) need for invasive mechanical ventilation, (2) shock (i.e., vasopressors needed), (3) renal replacement therapy, and (4) death.

The secondary endpoints were the plasma levels of biomarkers: highly sensitive troponin, NT-pro-BNP, interleukin-6 (IL-6), sRAGE, VEGF-A and GRP78.

### 2.3. Blood Samples

Whole blood was collected on ethylenediaminetetraacetic acid (EDTA) tubes within the first 24 h after ICU admission, using a venous or arterial catheter, depending on their availability. Tubes were immediately centrifugated for 15 min (2000 G, 4 °C) and plasma was frozen at −20 °C for a maximum of 2 weeks before being frozen at −80 °C until final assays.

### 2.4. Assays

The present study is an ancillary study of the COVID-THELIUM cohort. Assays were performed using an Enzyme-Linked Immunosorbent Assay for sRAGE, VEGF-A (Invitrogen, ThermoFisher Scientific, Waltham, MA, USA) and GRP78 (Enzo Life Sciences, Villeurbanne, France), and using an Electro-Chemiluminescence Immunoassay for IL-6 (Roche Diagnostics, Meylan, France).

### 2.5. Data and Statistical Analysis

The primary objective of the study was to evaluate the association between the plasma levels of the different biomarkers at ICU admission and the occurrence of a worsening towards a “critical” condition during the 28 first days, as previously defined above. Severe patients who did not fulfil the “critical” criteria were defined as “not critical”.

Because of the non-normal distribution of most of the data, as observed using a D’Agostino test, the results are expressed as medians with interquartile ranges (IQR) for quantitative data, and as absolute numbers and percentages (*n*, %) for qualitative data. In cases of missing data, no value was imputed.

First, we compared “critical” and “not critical” patients using a Mann–Whitney two-tailed test for unpaired values. A χ^2^ test (or Fisher’s exact test) was used for categorical data. Statistical significance was assessed using an alpha level of 0.05.

Then, we assessed the association between worsening towards a critical condition and the plasma levels of biomarkers by considering clinical characteristics of interest at ICU admission that are known to be early predictors, including age, body mass index (BMI), Simplified Acute Physiology Score II (SAPS-II), and treatment with a corticosteroid or anticoagulant at admission. The predictor variables, with a *p* value of <0.2 in this univariate analysis, were included in the model, using a logistic regression model with a stepwise selection process. *p* values were computed using Wald’s method. The discrimination and the calibration of the model were internally validated by means of the Hosmer–Lemeshow goodness of fit test and the area under the operating characteristic curve (AUC) to assess model discrimination.

Thereafter, the predictive values and corresponding 95% confidence intervals of the significant factors identified in the multivariable analysis were assessed by the area under the receiver operating characteristic (ROC) curve.

Optimal cut-off values, defined as the best combination of sensitivity and specificity, were determined using the Youden index.

All statistical tests were two-sided, and the 0.05 probability level was used to establish statistical significance. The statistical analyses were performed by means of the statistical software SAS (version 9.4; SAS Institute, Cary, NC, USA).

## 3. Results

A total of 100 patients were included between May and October 2020. Among them, two were excluded because of a wrong diagnosis of COVID-19 pneumonia, resulting in a total of 98 analyzed patients. Finally, 59 patients remained stable, whereas 39 patients progressed to a critical condition. Among these 39 patients, 29 required vasopressors, 37 required invasive mechanical ventilation, three required renal replacement therapy, and 13 died. The clinical and biological characteristics of interest at ICU admission are presented in Table 1. Briefly, at admission, patients worsening towards a critical condition were older, and presented higher SAPS-2 and SOFA severity scores, and lower ROX indexes and PaO_2_/FiO_2_ ratios. The median duration of invasive mechanical ventilation was 15 [8–21] days. Critical patients presented worse kidney function. No difference was observed concerning the specific treatment for COVID-19 disease at ICU admission.

Concerning biomarkers, critical patients presented higher plasma levels of troponin (16.5 [9–39] vs. 10 [6–18] pg/mL, *p* = 0.003), NT-pro-BNP (320.5 [118–1446] vs. 169 [63–366] pg/mL, *p* = 0.009), sRAGE (626 [449.5–1043] vs. 227 [137–404] pg/mL, *p* < 0.0001) and IL-6 (43 [15–112] vs. 11 [5–20] pg/mL, *p* < 0.0001). No difference was observed for VEGF-A (197 [75–395] vs. 169 [64.5-286] pg/mL, *p* = 0.6) and GRP78 (1596 [1190–2014] vs. 1569 [1187–1988] pg/mL, *p* = 0.97). The results are presented in Figure 1.

The univariable analyses showed an association between the occurrence of a critical condition and age (*p* = 0.0002), SAPS-II score (*p* < 0.0001), plasma levels of creatinine (*p* = 0.007), troponin T (*p* = 0.006) and sRAGE (*p* < 0.0001) (Table 2). Then, a multivariable analysis was performed including these variables and NT-pro-BNP and IL-6 plasma levels (*p* < 0.2). Only sRAGE plasma levels and age were significantly and independently associated with the occurrence of a critical condition within the first 28 days of ICU stay, with an odds-ratio of 1.03 [1.01–1.05] per 10 pg/mL of sRAGE and an odds ratio of 1.7 [1.2–2.4] per five years of age. Evaluation of the multivariate prediction model showed an AUC range from 0.86 to 0.98 and a *p* value for the Hosmer–Lemeshow (HL) goodness of fit of 0.359. This indicates good discrimination (i.e., separating “critical” patients from “not critical” patients) and calibration (the model fits the data well).

A ROC curve for sRAGE was elaborated for the prediction of worsening condition and showed an AUC of 0.82 [0.74–0.91] (Figure 2). A cut-off value of 449.5 pg/mL predicted worsening with a specificity of 80 and a sensitivity of 77% (Youden index), resulting in a positive predictive value of 71% and a negative predictive value of 84% in this cohort. A more sensitive cut-off value of 202 pg/mL predicted the worsening with a sensitivity of 97% and a specificity of 42%, resulting in a positive predictive value of 53% and a highly negative predictive value of 96% in this cohort.

## 4. Discussion

In this cohort study, we demonstrated that the soluble form of RAGE was independently associated with a worsening of the vital status of patients admitted to the ICU for severe COVID-19 pneumonia, in contrast to the other explored biomarkers. RAGE is highly expressed in alveolar type 1 cells and is a key player in pro-inflammatory pathways, as demonstrated by the increased alveolar level in experimental models of acute lung injury [24,25,26]. Soluble forms of RAGE are produced both from the proteolytic cleavage of the membrane receptor and from endogenous production by alternative splicing [27]. Elevated plasma levels were observed during pulmonary inflammation [28]. In a meta-analysis including 746 patients from eight studies, sRAGE was independently associated with 90-day mortality during acute respiratory distress syndrome (ARDS), highlighting the prognostic relevance of this marker [29]. Pathophysiological changes in RAGE pathways during COVID-19 disease make this marker even more interesting in this specific setting. Indeed, beyond the acute pulmonary damages related to the virus penetration in pneumocytes, a direct crosstalk may exist between RAGE and SARS-CoV-2 infection. The spike protein interacts with angiotensin-converting enzyme 2 (ACE2) as a receptor for viral penetration. A side effect of this interaction is the downregulation of ACE2 and the subsequent accumulation of angiotensin II within cells. This induces inflammatory activation through angiotensin receptor type 1, with the production of inflammatory markers including HMGB1. This latter factor may amplify inflammation by the activation of RAGE pathways [7]. This specific mechanism of action may explain the associated comorbidities described as worsening the outcomes of COVID-19 patients, such as hypertension, obesity, diabetes, or aging, all situations where the RAGE axis is often disturbed, notably by the higher production of AGE, HMGB1 or S100 ligands [30]. In a recent cohort study of hospitalized non-ICU COVID-19 patients, higher sRAGE plasma levels were associated with higher mortality [31]. In another cohort study, Lim et al. described sRAGE plasma levels in 164 COVID-19 patients of varying severity (including 32 patients of severe status, based on a definition similar to the one we used) [32]. RAGE plasma levels were correlated with the severity of the disease and predicted the need for invasive mechanical ventilation with good accuracy. In our study, we focused only on critically ill patients with respiratory failure, with 94% of patients treated with HFNC or NIV at admission. Soluble RAGE appeared to be a good prognostic marker of worsening towards multi-organ failure or death, and thus may be useful to identify patients at risk of complications, and who may be eligible for reinforced monitoring in the ICU, rather than intermediate care. In comparison, IL-6 plasma levels within 24 h of ICU admission presented an AUC of 0.62 in a cohort of critically ill COVID-19 patients, whereas sRAGE in our cohort presented a higher AUC of 0.82, suggesting a better prognosis performance. These potentially novel findings confirm, in a larger population size, that plasma sRAGE may be a promising biomarker in COVID-19 pneumonia. Although the measurement of plasma sRAGE is currently reserved for research purposes, our data, in combination with previous studies, argue for the development of the routine laboratory analysis of sRAGE in order to identify “at-risk” patients and to refer them as soon as possible to the most appropriate unit.

In our study, no association was observed regarding GRP78 and ICU outcome. GRP78 is an essential chaperone protein involved in the unfolded protein response (UPR) and its expression is regulated by activating transcription factor 4, which is activated in stress situations where there is a risk of protein misfolding, such as oxidative stress, anoxia/hypoxia, acute inflammation and amino acid deprivation [16]. UPR consists of a decrease in protein synthesis and an enhanced capacity of endoplasmic reticulum protein folding, in order to maintain cellular homeostasis. Imbalance in this process may lead to cellular apoptosis [33]. Some studies have explored the level of expression of GRP78 during COVID-19 disease. In an autopsy study, a higher expression of GRP78 was observed in the pneumocytes and alveolar macrophages of a COVID-19 patient with pneumonia, compared with control patients [15]. In a small cohort study, plasma levels of GRP78 were shown to be significantly higher in COVID-19 patients in comparison with control patients. Nevertheless, no difference was observed between COVID-19 patients with respiratory failure compared with asymptomatic patients, suggesting, similarly to our results, that GRP78 is elevated in COVID-19 patients but is not associated with the severity of disease [34]. Several hypotheses may be raised to explain these results. First, most severely ill COVID-19 patients mainly present isolated respiratory failure, which may explain the low sensitivity in the plasma of GRP78 as a prognostic biomarker, even though UPR signaling may be of great magnitude in the lungs. Indeed, very few patients presented vascular or renal failure at admission in our study. Secondly, ischemia/reperfusion and reactive oxygen species (ROS) are important determinants of ER stress and UPR activation. Contrary to cardiac surgery or sepsis, where these phenomena are frequent and generalized to the whole organism, ROS production during COVID-19 disease seems to be mainly localized to the lungs [35,36,37,38]. Thus, in our study, GRP78 does not appear as a biomarker of severity in a COVID-19 ICU population.

Another biomarker we evaluated was VEGF-A, with no difference observed according to the prognosis of patients. VEGF-A is a growth factor of importance in the regulation of endothelial function, and of microcirculation in general, notably because of its action through the type 1 and 2 VEGF receptors. The activation of these receptors induces many phosphorylation processes, and notably the destabilization of intercellular junctions, leading to vascular permeability [18]. Many studies have demonstrated increased plasma levels of VEGF-A during sepsis, a pathology where microcirculation is highly disturbed [21,39]. Microcirculation is also challenged during the COVID-19 disease course, as demonstrated both by elevated plasma levels of angiopoietin 2 and by the in vivo demonstration of a reduction in small-vessel density [1,40]. Nevertheless, and despite plasma levels similar to those observed during sepsis, VEGF-A was not able to discriminate patients with worse outcomes in our study. This absence of discrimination of VEGF-A, according to the prognosis of COVID-19 patients, has already been described in small cohort studies, contrary to other endothelial biomarkers, such as angiopoietin-2 or the Intercellular Adhesion Molecule type 1 (ICAM-1) [1,41,42]. Thus, we confirmed the lack of interest of VEGF-A in a larger population.

As evoked for GRP78, the inability of GRP78 and VEGF-A to discriminate patients with a poor prognosis in our study may be related to the relatively few cases with systemic insults. Thus, the measurement of these biomarkers at admission, when the patients only presented a lung affliction, may have been too early in the course of the disease. Because RAGE is essentially expressed in the lung, and because of the existence of a soluble form, this could explain why this marker may be efficient to discriminate patients as early as admission.

We also demonstrated in this study that patients who evolved towards a critical condition presented higher plasma levels of troponin and NT-proBNP at ICU admission. Although significant, these levels were only slightly higher than those observed in healthy patients and were not independently associated with a worsening condition. Moreover, we only observed one patient in the whole cohort with confirmed cardiogenic shock, but we could not rule out cases of sub-clinical myocarditis. Indeed, higher levels of troponin have been described in up to 25% of hospitalized COVID-19 patients, but with only 7% presenting true myocardial suffering. Moreover, patients with higher troponin levels were more likely to require ICU admission [43]. Taken together, troponin appears to be associated with the severity of SARS-CoV-2 infection, but not with patient prognosis.

We observed a significant and independent association with worsening condition in elderly patients, as previously published [44]. Indeed, elderly patients frequently present comorbidities that put them at risk of severe presentations of COVID-19, including hypertension, diabetes, obesity, pulmonary or renal chronic diseases and dementia [45]. Other clinical or biological markers may be of interest in identifying patients at risk of a worsening condition. Among them, the alteration of hemostatic processes may be of interest, predominantly because of the higher incidence of thrombotic events during COVID-19 disease. We previously described the alteration in the coagulation process in our cohort, with a particular role identified for immature platelets and thrombin generation assay, whereas d-dimers presented discouraging results [3,46]. On the other hand, clinical parameters such as the ROX index, which quantifies the severity of respiratory failure under HFNC, may predict the respiratory worsening of patients [47]. Thus, it seems to us wise to consider the triage of patients eligible for admission to the ICU according to a set of clinical and biological parameters, among which age, severity of respiratory failure and plasma levels of sRAGE may play a promising role.

Our study presents several limitations. First, despite a relatively important number of patients, we cannot rule out that the cohort is underpowered to demonstrate independent associations between certain biomarkers (in particular, troponin and NT-pro-BNP) and outcomes. Secondly, and as discussed above, the measurements were performed very early in the course of the ICU stay. While it is of interest to identify patients “at-risk” of worsening as early as possible, GRP78 and VEGF-A levels may have been different between the two groups later in the ICU stay, and especially when patients developed shock or kidney failure. Thus, even if these biomarkers cannot be used as early biomarkers of prognosis, it is very plausible that their pathological pathways are disturbed later in course of the disease.

## 5. Conclusions

Our study suggested that sRAGE monitoring may be useful to identify COVID-19 patients at risk of worsening towards a critical and life-threatening condition. The routine measurement of this marker may help practitioners in the triage process, and particularly when hospital resources are strained during epidemic waves.

## Figures and Tables

**Figure 1 jcm-11-04571-f001:**
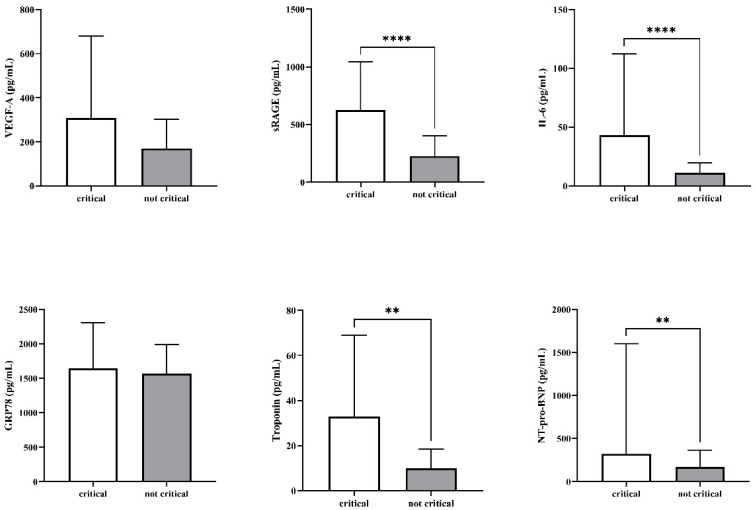
Plasma levels of the different biomarkers at ICU admission with regard to worsening towards a critical condition (critical group) or not (not critical group). ** *p* < 0.01, **** *p* < 0.0001.

**Figure 2 jcm-11-04571-f002:**
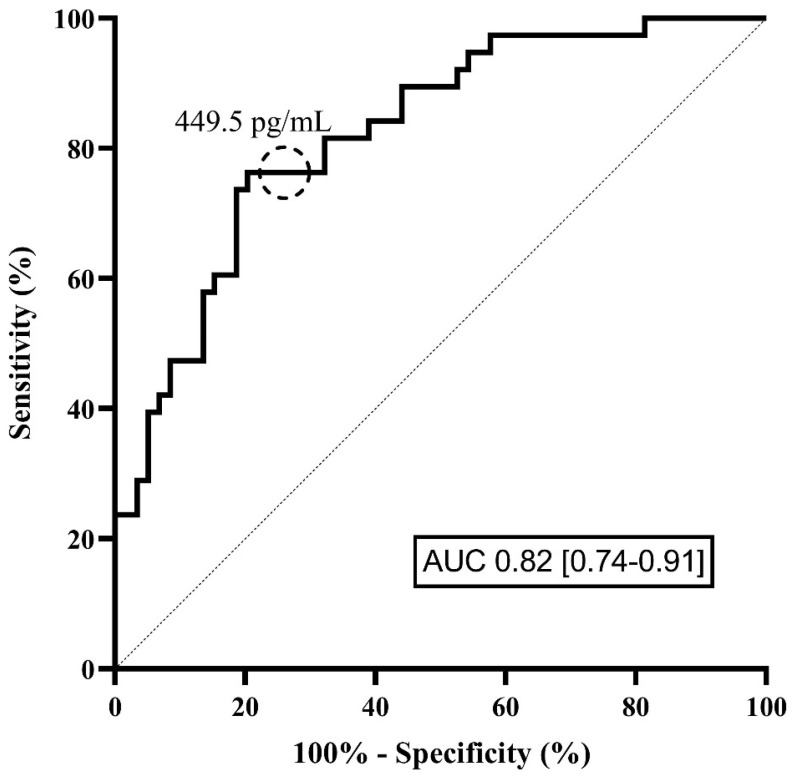
Receiver operating characteristic (ROC) curve for the prediction of the worsening of patients towards a critical condition during their ICU stay, according to the plasma level of sRAGE at ICU admission. The cut-off value of 449.5 pg/mL predicted worsening with a sensitivity of 77% and a specificity of 80%, as identified using the Youden index.

**Table 1 jcm-11-04571-t001:** Clinical and biological characteristics at admission to the ICU.

Parameters	All (*n* = 98)	Not Critical (*n* = 59)	Critical (*n* = 39)	*p*-Value
Age (years)	67 [58–73]	61 [53–69]	72 [64–75]	<0.0001
Male (*n*, %)	68 (69.4)	40 (67.8)	28 (71.8)	0.6742
Body mass index (kg/m^2^)	29.2 [25.3–33.9]	29.9 [26.1–35.2]	29 [25.1–32.4]	0.38
Obesity (BMI ≥ 30 kg/m^2^) (*n*, %) (*n* = 58/39)	46 (47.4)	29 (50)	17 (40.2)	0.54
Underlying comorbidity (*n*, %)				
Chronic pulmonary disease	9 (9.2)	5 (8.5)	4 (10.3)	1
Asthma	15 (15.3)	12 (20.3)	3 (7.7)	0.09
Diabetes	42 (42.9)	24 (40.7)	18 (46.2)	0.6
Hypertension	60 (61.2)	35 (59.3)	25 (64.1)	0.63
Peripheral arterial disease	3 (3.1)	1 (1.7)	2 (5.1)	0.56
Coronaropathy	9 (9.2)	4 (6.8)	5 (12.8)	0.48
Smoking	4 (4.1)	2 (3.4)	2 (5.1)	/
Active neoplasia	9 (9.2)	7 (11.9)	2 (5.1)	0.31
COVID-19 related treatment at admission (*n*, %)				
Corticosteroid	90 (91.8)	56 (94.9)	34 (87.2)	0.26
Remdesivir	17 (17.4)	13 (22)	4 (10.3)	0.13
Lopinavir/ritonavir	7 (7.1)	4 (6.8)	3 (7.7)	1
Tocilizumab	1 (1.0)	1 (1.7)	0 (0)	/
Anticoagulation therapy (*n*, %)	90 (91.8)	55 (93.2)	35 (89.7)	0.71
Prophylactic intensity	8 (8.2)	4 (6.7)	4 (10.5)	
Intermediate intensity	61 (62.2)	42 (70)	19 (50.0)	
Therapeutic intensity	22 (22.4)	10 (16.7)	12 (31.6)	
ICU transfer since the onset of symptoms (days)	9 [6–10]	9 [7–10]	7 [4–10]	0.06
SAPS II score	32.5 [25–40]	27 [22–35]	40 [34–52]	<0.0001
SOFA score	3 [1–4]	2 [1–3]	4 [2–6]	<0.0001
Non-invasive respiratory support (HFNC or NIV) (*n*, %)	92 (93.9)	59 (98.3)	33 (86.8)	0.8
PaO_2_/FiO_2_ ratio (*n* =)	135 [95–165]	144 [117–168]	115.5 [78–159]	0.026
ROX index (*n* = 71)	6.5 [5.4–8.8]	7.5 [6.0–10.3]	5.4 [4.4–6.2]	<0.0001
Biological parameters				
Creatinine (µmol/L)	75 [55–101]	65 [52–81]	94 [76–156]	0.003
Urea (mmol/L)	7 [4.9–9.6]	5.7 [4.5–8]	8.9 [5.5–17.2]	0.0007
Hemoglobin (g/dL)	12.6 [11.4–13.7]	12.6 [11.6–13.7]	12.6 [11.2–13.6]	0.59
Platelets (G/L)	228.5 [160–287]	236 [179–303]	197 [138–253]	0.02
Leukocytes (G/L)	7.6 [6.0–11.3]	7.5 [5.8–10.2]	8.4 [5.9–12.1]	0.5
C-reactive protein (mg/L) (*n* = 59/37)	112 [72.5–187.5]	122 [76–186]	102 [63–189]	0.82
Lactatemia (mmol/L) (*n* = 57/39)	1.3 [1.0–1.7]	1.3 [0.9–1.6]	1.4 [1–1.7]	0.11
Glutamic–pyruvic transaminase (U/L)	45 [34–68]	44 [34–60]	50 [34–72]	0.2
Glutamic–oxaloacetic transaminase (U/L)	38 [24–65]	43 [29–68]	31 [23–58]	0.1

BMI: body mass index; ICU: intensive care unit; SAPS: Simplified Acute Physiology Score; SOFA: Sequential Organ Failure Assessment; NT-pro-BNP: N-terminal pro-brain natriuretic peptide.

**Table 2 jcm-11-04571-t002:** Associations between clinical and biological variables at ICU admission and worsening towards critical condition during ICU stay.

Variable	Univariable Analysis		Multivariable Analysis		Multivariable Analysis with a Stepwise Selection Process	
	OR [95% CI]	OR [95% CI]	OR [95% CI]	*p*	OR [95% CI]	*p*
**Age (per 5 years)**	**1.63 [1.26–2.10]**	**0.0002**	1.6 [1.04–2.5]	0.03	1.7 [1.2–2.4]	0.04
Male gender	1.21 [0.50–2.93]	0.67				
BMI (per kg.m^−2^)	0.97 [0.91–1.04]	0.40				
**SAPS-II score**	**1.10 [1.05–1.15]**	**<0.0001**	1.06 [0.99–1.13]	0.10		
No corticosteroid	2.75 [0.62–12.2]	0.19				
No anticoagulation	1.57 [0.37–6.69]	0.54				
**Creatinine (per µmol/L)**	**1.02 [1.00–1.03]**	**0.007**	0.99 [0.98–1.01]	0.4		
**Troponin (per pg/mL)**	**1.05 [1.01–1.08]**	**0.006**	1.02 [0.98–1.07]	0.25		
**NT-pro-BNP (per 10 pg/mL)**	**1.002 [0.99–1.005]**	**0.12**	1.00 [0.99–1.004]	0.85		
**sRAGE (per 10 pg/mL)**	**1.03 [1.02–1.05]**	**<0.0001**	**1.04 [1.01–1.06]**	**0.005**	1.03 [1.01–1.05]	0.001
**IL-6 (per 5 pg/mL)**	**1.02 [0.99–1.05]**	**0.1**	1.02 [0.99–1.05]	0.13		
VEGF (per 5 pg/mL)	0.99 [0.99–1.004]	0.81				
**GRP78 (per 10 ng/mL)**	**0.99 [0.99–1.01]**	**0.83**	0.99 [0.99–1.01]	0.87		

Bold variables with *p* values < 0.2 are included in logistic regression model analysis. BNP: brain natriuretic peptide; GRP-78: glucose-related protein 78; IL-6: interleukin-6; SAPS: Simplified Acute Physiology Score; sRAGE: soluble receptor for advanced glycation end product; VEGF: vascular endothelial growth factor. Analyses were performed using a logistic regression model to explain the occurrence of worsening towards a critical condition during the first 28 days of ICU stay (invasive mechanical ventilation, renal replacement therapy, vasopressor therapy and/or death).

## Data Availability

Data are available upon reasonable request from the corresponding author.

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
