# Peer review of "Soluble RAGE as a Prognostic Marker of Worsening in Patients Admitted to the ICU for COVID-19 Pneumonia: A Prospective Cohort Study"

_jcm, 2022, doi:10.3390/jcm11154571_

Round 1
Reviewer 1 Report
Besnier and colleagues suggested that sRAGE monitoring may be useful to identify severe Covid-19 patients at-risk of evolving towards a critical and life-threatening condition. I have few concers which are listed below
1. Unit of GRP78 is not clear. In line 170 -171, Unit of GRP78 is pg/mL while in fig 1 unit is ng/mL.
2. Expand the introduction and discussion.
3. Need the English revision
4. In addition to sRAGE,InterLeukine-6, troponin T and NT-pro-BNP, VEGF-A and GRP78, C-reactive protein (CRP), D-dimer are also important parameters to measure. Why did not authors measured these?
5. Refrence 17 is not complete
Author Response
- Unit of GRP78 is not clear. In line 170 -171, Unit of GRP78 is pg/mL while in fig 1 unit is ng/mL.
We are sorry for the mistake. The correct unit was pg/mL. It has been corrected in figure 1
- Expand the introduction and discussion.
We expanded the introduction by discussing the limitation of IL-6 has a prognosis marker and why we think it is of interest to explore the “second wave” biomarkers (biomarkers resulting from the primary inflammation). We also discussed why we explored VEGF-A in the introduction rather than in the material and methods
We expanded the discussion by comparing our results for sRAGE with the results for IL-6 in a recent publication with a similar design. We also discussed our results concerning age and Rox index, as asked by another reviewer.
- Need the English revision
The manuscript has been edited through the English editing service from mdpi
- In addition to sRAGE,InterLeukine-6, troponin T and NT-pro-BNP, VEGF-A and GRP78, C-reactive protein (CRP), D-dimer are also important parameters to measure. Why did not authors measured these?
We have already published our first results of our cohort, concerning hemostasis disorder from this cohort in the Journal of Clinical Medicine (ref 3), including the analysis of D-dimers. The current results come from the second analysis, realized later. It is therefore note possible to include these results in this new publication. We discuss this point line 329
- Refrence 17 is not complete
We completed the reference by adding the page number. This reference is not indexed and thus does not present a doi.
Reviewer 2 Report
IN the manuscript jcm-1824272- The authors aimed to recognize the patients at risk of the clinical worsening from a “severe” towards a “critical” condition within 28 days after ICU admission.
In the study design, the authors are mentioning trial registration number NCT04357847 Assessment of Endothelial and Haemostatic Changes During Severe SARS-CoV-2 Infection (Covid-Thelium).
The primary endpoint of the study NCT04357847 „Assessment of Endothelial and Haemostatic Changes During Severe SARS-CoV-2 Infection (Covid-Thelium)“ are changes in
Association of InterCellular Adhesion Molecule-1 plasma level with 28 days mortality – not investigated in this study.
Secondary Outcomes were:
Association of Endothelin-1 plasma level with 28 days mortality
Association of Vascular Endothelial Growth Factor A plasma level with 28 days mortality
Association of soluble Vascular Endothelial Growth Factor Receptor type 1 with 28 days mortality
Association of syndecan -1 plasma level with 28 days mortality
Association of D-dimers plasma levels with thrombotic events
Association of von Willebrandt Factor with thrombotic events
Association of Viscoelastic testing with thrombotic events
Only few endpoints that were registered were published in the previous publication (ref. 3) it was focused to some coagulation changes.
Endpoints of this paper are:
- Primary outcome: clinical worsening from a “severe” towards a “critical” condition within 28 days after ICU admission
- The secondary endpoints were the plasma levels of biomarkers: highly sensitive troponin T, NT-pro-BNP, Interleukine-6 (IL-6), sRAGE and GRP78, Vascular Endothelial Growth Factor-A (VEGF-A)
It is obvious that this is a completely different study, and the population is also not the same, although some blood samples may have been taken from patients from a previous study population. Furthermore, in the NCT04357847 all endpoints are focused on the association of some laboratory markers with 28-day mortality, while this study investigates the clinical worsening from a "severe" towards a "critical". I, therefore, recommend just mentioning previously published study in the reference list without the NCT04357847. NCT may be obtained separately for this study and must list the outcomes investigated here.
Please specify – whether 13 patients from the previous publication were included in this study or 13 new patients were added - it is not clear.
The primary endpoint of the study was the clinical worsening from a “severe” towards a “critical” condition. The primary outcome was not discussed, although several parameters were identified in Table 1 as important and statistically significant. It seems that the authors prefer to comment on laboratory markers that are not significant rather than clinical markers that are significant. Despite authors are stating that it” is essential to correctly refer patients to adequate structure and spare limited resources” they do not investigate neither healthcare structure nor resources.
Minor mistakes to be corrected:
p. 3 ln 115 - please change being frozen at 20°C and 80°C to -20 and -80°C.
Who is XX person in the Author Contributions?
Multiple minor typos are present like:
-plasma levels where associated.
- so 94% of patients treated with HFNC
- failure compare with asymptomatic etc.
Author Response
It is obvious that this is a completely different study, and the population is also not the same, although some blood samples may have been taken from patients from a previous study population. Furthermore, in the NCT04357847 all endpoints are focused on the association of some laboratory markers with 28-day mortality, while this study investigates the clinical worsening from a "severe" towards a "critical". I, therefore, recommend just mentioning previously published study in the reference list without the NCT04357847. NCT may be obtained separately for this study and must list the outcomes investigated here.
We thank the reviewer for these precisions. We recognized that the endpoints of the study have changed from the initial methodology published in clinicaltrials.gov. Nevertheless, it is still the same cohort study of 100 ICU Covid-19 patients, with blood samples taken within 24 h. We changed the biomarkers that we explored because of the evolution of the literature and the previously published data on ICAM-1 at the time of the assays, and we also modified the primary endpoint into a composite endpoint because of the publication of the NMA-Covid Initiative Network definitions as described in the material and methods section. Indeed, the study was design after the first epidemic wave where mortality was very high. Hopefully we observed a reduction in mortality during the following waves, and thus it appears than exploration of intense ICU stay, with organ supports, may be also an important point to improve, and that exploring only mortality will be limited.
Because of these changes, and accorsding to the suggestion, we suppressed the reference to the NCT number. Nevertheless, regestering a posteriori to a new NCT is not possible because of the prospective nature of the cohort.Nevertheless, because this study is not an interventional trial, and according to the FDA recommendations (and to the French law on trials), a NCT is not required for cohort studies. We suggest to suppress the NCT without replacement.
Please specify – whether 13 patients from the previous publication were included in this study or 13 new patients were added - it is not clear.
The current study and the previous publication explored different objectives in the same cohort of 100 severe COVID-19 patients admitted to the ICU. Over the 100 patients, 2 patients were not analyzed because of a wrong diagnosis of Covid-19 (as explained in the first lines of the results section), resulting in 98 analyzed patients in the present manuscript. In the first publication, only 85 patients were analyzed because of technical issues in the processing of the coagulation tests, for 13 patients.
Nevertheless, we suggest to suppress the reference to the first publication concerning the 13 "new patients", because it does not add informations to the reader and does not change the scientific interest or reliability of the present study.
The primary endpoint of the study was the clinical worsening from a “severe” towards a “critical” condition. The primary outcome was not discussed, although several parameters were identified in Table 1 as important and statistically significant. It seems that the authors prefer to comment on laboratory markers that are not significant rather than clinical markers that are significant. Despite authors are stating that it” is essential to correctly refer patients to adequate structure and spare limited resources” they do not investigate neither healthcare structure nor resources.
We added a discussion of the clinically significant findings (age and Rox index) line 325. We did not discuss the role of severity scores because these scores are mainly used to adjust the multivariate model (we arbitrary chose to include SAPS-2 over SOFA score in the model because of its multiparametric proprieties). We added a sentence to suggest that the reader take in consideration sRAGE but also age, respiratory status and other clinical parameters of interest for triage of patients.
We effectively did not investigate the healthcare structure because our study took place in a unique facility.
Minor mistakes to be corrected:
- 3 ln 115 - please change being frozen at 20°C and 80°C to -20 and -80°C.
This has been corrected
Who is XX person in the Author Contributions?
Reference to XX has been suppressed
Multiple minor typos are present like:
-plasma levels where associated.
This has be corrected
- so 94% of patients treated with HFNC
This has been corrected
- failure compare with asymptomatic etc.
The manuscript has been edited by the english editing service from mdpi
Round 2
Reviewer 2 Report
In this version of the text, the authors have clarified most of the ambiguities from the previous version. Since the study has been completed, further improvements are not possible. The text is now quite readable and acceptable for publishing.